# Modeling and Optimizing the Crystal Violet Dye Adsorption on Kaolinite Mixed with Cellulose Waste Red Bean Peels: Insights into the Kinetic, Isothermal, Thermodynamic, and Mechanistic Study

**DOI:** 10.3390/ma16114082

**Published:** 2023-05-30

**Authors:** Razika Mecheri, Ammar Zobeidi, Salem Atia, Salah Neghmouche Nacer, Alsamani A. M. Salih, Mhamed Benaissa, Djamel Ghernaout, Saleh Al Arni, Saad Ghareba, Noureddine Elboughdiri

**Affiliations:** 1Pollution & Waste Treatment Laboratory (PWTL), University of Ouargla, P.O. Box 511, Ouargla 30000, Algeria; 2Department of Chemistry, Faculty of Exact Sciences, University of El-Oued, P.O. Box 789, El-Oued 39000, Algeria; neghmouchenacer-salah@univ-eloued.dz; 3Chemical Engineering Department, College of Engineering, University of Ha’il, P.O. Box 2440, Ha’il 81441, Saudi Arabia; samani15@hotmail.com (A.A.M.S.); djamel_andalus@yahoo.fr (D.G.); s.alarni@uoh.edu.sa (S.A.A.); 4Department of Chemical Engineering, Faculty of Engineering, Al Neelain University, Khartoum 12702, Sudan; 5Chemical Engineering Department, Faculty of Engineering, University of Blida, Blida 09000, Algeria; 6Department of Chemical and Petroleum Engineering, University Elmergib, Al-Khums P.O. Box 40161, Libya; saad.ghareba@gmail.com; 7Chemical Engineering Process Department, National School of Engineers Gabes, University of Gabes, Gabes 6029, Tunisia

**Keywords:** kaolinite, red bean peels (RBPs), cellulosic waste, adsorption, crystal violet (CV), Box–Behnken design (BBD)

## Abstract

In this study, a new eco-friendly kaolinite–cellulose (Kaol/Cel) composite was prepared from waste red bean peels (*Phaseolus vulgaris*) as a source of cellulose to serve as a promising and effective adsorbent for the removal of crystal violet (CV) dye from aqueous solutions. Its characteristics were investigated through the use of X-ray diffraction, Fourier-transform infrared spectroscopy, scanning electron microscopy, energy-dispersive X-ray spectroscopy, and zero-point of charge (pH_pzc_). The Box–Behnken design was used to improve CV adsorption on the composite by testing its primary affecting factors: loading Cel into the composite matrix of Kaol (A: 0–50%), adsorbent dosage (B: 0.02–0.05 g), pH (C: 4–10), temperature (D: 30–60 °C), and duration (E: 5–60 min). The significant interactions with the greatest CV elimination efficiency (99.86%) are as follows: BC (adsorbent dose vs. pH) and BD (adsorbent dose vs. temperature) at optimum parameters (A: 25%, B: 0.05 g, C: 10, D: 45 °C, and E: 17.5 min) for which the CV’s best adsorption capacity (294.12 mg/g) was recorded. The Freundlich and pseudo-second-order kinetic models were the best isotherm and kinetic models fitting our results. Furthermore, the study investigated the mechanisms responsible for eliminating CV by utilizing Kaol/Cel–25. It detected multiple types of associations, including electrostatic, *n*-π, dipole–dipole, hydrogen bonding interactions, and Yoshida hydrogen bonding. These findings suggest that Kaol/Cel could be a promising starting material for developing a highly efficient adsorbent that can remove cationic dyes from aqueous environments.

## 1. Introduction

Today, the planet faces serious pollution issues, a global concern for the natural ecosystem. Much dye-containing wastewater relating to human activities is discharged without treatment, seriously affecting the environment and drinking water quality [1]. As contaminants in North African nations, including Algeria, dyes are dumped into the water without treatment [2]. Dyes can be divided into three categories according to their core structure: cationic dyes (basic), anionic dyes (acidic and reactive), and nonionic dyes (disperse) [3]. Organic cationic dyes, such as crystal violet (CV), are more toxic to humans and other living organisms than anionic ones [4]. Reducing or eliminating the dye content of wastewater before discharging it into the environment is a way to handle this significant ecological problem [5]. Several techniques for removing anionic dyes from wastewater have been presented, including ion exchange, membrane filtration, and irradiation, as well as chemical approaches such as oxidation, coagulation, zonation, photochemical and electrochemical destruction. Biological methods such as aerobic and anaerobic microbial degradation and photochemical treatments have also been proposed [6,7,8]. However, the adsorption method is still the method of choice for dye removal because it has several advantages over other methods other forms do not have. These advantages include a simple design, a broad pH, efficacy even at low pollutant concentrations, selectivity, the ability to be regenerated, and the design is easy to operate in moderate conditions [9,10]. Many researchers have focused on adsorption technology employing cheap adsorbents [11].

Clays and their composites have recently emerged as effective adsorbents for dye-containing water. This is due to the clays’ high microporosity, high stability, large area of a specific surface, excellent swelling ability, high cation-exchange capacity (CEC), and environmental safety [12,13]. It has been discovered that clays and the composites made from them are excellent in removing dye from water while being quite inexpensive. Kaolinite is a well-known example of a layered substance or clay, which may be found at [14]. Kaolinite is an aluminum silicate clay with a 1:1 dioctahedral layered structure consisting of a single SiO_4_ tetrahedral layer and an Al(OH)_6_ octahedral sheet in each layer. These layers are interconnected by an O-H-O bond. Kaolinite exhibits adsorption properties and is therefore used for eliminating various toxins from wastewater [15]. However, compared to illite, chitosan, activated carbon, or zeolites, raw kaolinite has a lower adsorption capacity and CEC (3–15 mEq/100 g) [16]. Adsorbing pollutants on natural minerals decreases capacity after a few cycles [9,16]. Several techniques have been used to modify clay, such as thermal, acid/base activation, intercalation, and metal oxide coatings. These modifications have led to the development of exceptional adsorbents that exhibit improved sorption capabilities by increasing surface area, pore size and volume, and surface binding sites [17]. These modifications have led to the development of special adsorbents that exhibit improved adsorption performance through increased surface area, pore size, volume, and surface binding sites [17].

Recently, biopolymer/clay bio-nanocomposites have gained attention as appealing materials for removing pollutants from contaminated water thanks to their biodegradability, biocompatibility, large surface area, and remarkable sorption performance [18]. Among the several varieties of biopolymers (such as chitosan, alginate, and starch), cellulose is ubiquitous and renewable and has abundant natural availability. As a result, it is drawing a lot of attention. Further, it is biocompatible, biodegradable, has high mechanical strength, is environmentally efficient, and does not contain toxic elements. Cellulose is made up of linear polysaccharides that are made up of two anhydroglucose rings that are associated with several β-1,4 glycosidic bonds [19]. Nevertheless, cellulose’s chemical composition makes it more likely to absorb cationic than anionic dyes. The reason for this is that the lignocellulose composition comprises multiple anions (such as hydroxyl (-OH) and carbonyl (-CO-) groups) serving as active sites. Moreover, it usually has a negative surface charge [20]. In this area, cellulose-based materials [21], hydrogels [22], and agricultural and other natural plant-derived wastes it is predominantly cellulose (40–50%), such as rice husk, sawdust, corn stalks, orange peel, wheat straw, cotton waste, banana waste, coffee waste, date palm tree spathe sheath and many other resources [6,23], and have been presented for the effective separation numerous pollutants including cationic dyes from wastewater. Several inorganic-organic hybrid composites are efficient adsorbents in a wide variety of experiments. These composites, which are used for eliminating different contaminants, i.e., cellulose/montmorillonite mesoporous beads [24], hydrogel based on cellulose and clay [25], kaolinite/biopolymer composites [26], cellulose@organically modified montmorillonite [27], cellulose/kaolinite–zeolite composite [28], are employed to remove different pollutants. They have shown promising efficiency for sequestering dyes. Because of the benefits offered by the materials described in the previous paragraph (Kaol, Cel), we decided to synthesize Kaol-Cel biopolymers as a recoverable adsorbent for removing CV dye from aquatic environments. This is the first cellulose-based compound extracted from red bean peels (*Phaseolus vulgaris*) (RBPs), which gives this study its innovative quality. To achieve maximum CV dye removal by Kaol/Cel, the experimental critical parameters needed to be improved and predicted using response surface techniques, included in Box–Behnken design (BBD). The investigation involved determining the synthesis conditions, including the loading of Cel into the Kaol biopolymeric matrix, as well as other crucial factors related to adsorption (such as adsorbent injection, pH, temperature, and residence period). Moreover, the study explored the kinetics, isotherms, and thermodynamic functions associated with the technique.

## 2. Materials and Methods

### 2.1. Material, Methods, and Instruments

RBPs were acquired from a local farm situated in El-Oued, Southeast Algeria. The samples underwent a thorough cleaning process to eliminate any dust or debris that may have been present on their surfaces. They were subsequently dried at 70 °C for 7–8 h and finely ground using a mixer grinder to ensure the samples were impurities-free. Raw kaolinite was obtained from Aougrout, Adrar, Southwest Algeria. The dye used in the study was CV (C_25_N_3_H_30_Cl, *λ*_max_ = 590 nm, molecular weight: 407.979 g/mol), purchased from Sigma-Aldrich. Chemical products, such as hydrogen peroxide (H_2_O_2_), hydrochloric acid (HCl), sodium hydroxide (NaOH), acetic acid (CH_3_COOH), sodium carbonate (Na_2_CO_3_), sodium hexametaphosphate (NaPO_3_)_6_, thiourea (SC(NH_2_)_2_), and sodium hypochlorite (NaOCl), were furnished by Merck (Darmstadt, Germany). The chemicals employed in this research were pure and of analytical quality. The characterization of Kaol-Cel– 25 was carried out using Fourier transform–infrared (FT-IR) analysis (Cary 600 series FT-IR spectrophotometer, Agilent technologies, Santa Clara, CA, USA [29]), X-ray diffraction (XRD) analysis (PANalytical X’Pert PRO diffractometer), scanning electron microscopy (SEM)-energy-dispersive X-ray (EDX) analysis (Hitachi, TM3030Plus, Tabletop Microscope), Brunauer–Emmett–Teller (BET) analysis (Micromeritics ASAP2020 system), and zero-point of charge (pH_pzc_) value was derived using the methodology indicated by Kaouah et al. [30].

### 2.2. Pretreatment of Raw Clay

To fraction the clay samples, the method followed here was founded on several processes already applied in previous works [31,32,33]: (i) A series of sieves were used in a cascade to perform preliminary pre-sieving on the unprocessed clay to obtain particle sizes of less than 5 μm; (ii) After that, it was ground into a powder gently and then rinsed with H_2_O_2_ (40 mL, 6% *w*/*v*) to remove any organic compounds; (iii) After that, the resulting solution was transferred to an Erlenmeyer (1000 mL) was added to the buffer solution of pH 4.8 (80 mL) (16 g sodium acetate and 10 mL of acetic acid); (iv) The resulting solution was transferred to a graduated tube with the addition of a dispersible component, sodium hexametaphosphate (NaPO_3_)_6_, and allowed to decant for 7 h and 45 min after that floating layer was collected at a depth of 10 cm to produce particles smaller than 2 μm. The granules that were acquired were treated numerous times employing deionized water and centrifuged; finally, (v) One glass slide served as the control and was not subjected to any treatment, while the other slide was subjected to heating at 105 °C for two hours to pick the suitable clay. Glass slides were immersed in a glycerin bath until they became saturated and were then analyzed using XRD to identify the presence of the clay mineral Kaolinite: Al_2_Si_2_O_5_(OH)_4_, which had the following composition: 39.7% alumina, 46.2% silica, and 13.8% water. The BET procedure was utilized for calculating the material’s specific area, which was 110.786 m^2^/g.

### 2.3. Extracting Cellulose from Red Bean Peels (RBPs)

The procedure shown in Figure 1 was used to isolate cellulose from the powder obtained from RBPs. The process, detailed in several publications [29,34,35], was slightly modified. Initially, 100 g of dried powder from RBPs was cooked in 3 L of hot water with a pH of 7 for 15–25 min and then filtered. The resulting mass was subjected to acid (HCl) pretreatment by mixing it twice with a 1 M HCl solution (500 mL) at 80–90 °C for 60 min; the residue was filtered and collected. For alkali (NaOH) pretreatment, the agitation of the residue occurred three times. In a 1.5 L solution of 1 M NaOH for 60 min at 80–85 °C and then collected. The filtered residue was then subjected to bleach treatment twice with a 4% (*w*/*v*) NaOCl solution (at pH = 5 adjusted with 10% (*v*/*v*) CH_3_COOH) for 60 min at 80–90 °C, resulting in white-colored cellulose. The cellulose was rinsed with hot deionized water multiple times until the filtrate pH was neutralized. Finally, the cellulose was freeze-dried for seven hours and ground using a mixer grinder. It was then kept at room temperature for further investigation.

### 2.4. Preparation of Kaol/Cel Composite

To create Kaol/Cel nanocomposites, the researchers followed a specific procedure [36,37]. Initially, acidification is conducted by dissolving 20 g of the pretreated kaolinite into 200 mL of sulfuric acid (H_2_SO_4_) (15% W) in the beaker glass and stirring for 4 h at 80 °C. Then, 10 g of activated kaolinite was mixed with 16 mL of (46% W) NaOH in ice water and subjected to magnetic stirring for six hours. To optimize the process according to BBD, the team loaded different ratios of Cel to the modified Kaol before dissolving it with 0.6 M NaOH and 1 M thiourea solutions. Finally, the resulting mixture was vigorously stirred at 80 °C for four hours and then oven-dried at 60 °C overnight.

### 2.5. Experimental Design

BBD is an effective method for optimizing processes since it conforms to a quadratic surface. As a result, it was included in the overall design of the experiment [38]. A total of forty-six separate tests were conducted to assess the impact of five primary independent variables on the removal of CV. To determine the ideal parameters and specify the experimental domain, preliminary tests were conducted. These studies focused on the following aspects: the loading of Cel into Kaol (A), the adsorbent injection (B), the pH (C), the residence period (D), and the temperature (E). The various experimental levels of independent variables and their respective codes are listed in Table 1. The BBD, as well as the statistical data, were examined with the help of software called Stat-Ease Design-Expert (Version 13.0). For the purposes of assessing and forecasting CV removal, the second-order polynomial model was used. Including all square terms, linear terms, and linear-by-linear interaction items, the quadratic response model can be defined as follows:(1)$$Y=\beta_{0}+\sum_{i=1}^{k}\beta_{i}\;\chi_{i}+\sum_{i=1}^{k}\beta_{ii}\;\chi_{i}^{2}+\sum_{i=1}^{k}\sum_{j=1}^{k}\beta_{ij}\;\chi_{i}\chi_{j}+\varepsilon$$
where *Y* represents the objective for optimizing the response, while *k* denotes the number of variables being considered. The indices *i* and *j* are used to represent the variable numbers, and *β*_0_ is the constant coefficient, with *β_i_* and *β_ii_* representing the linear and quadratic coefficients, respectively. The term *β_ij_* refers to the interaction coefficient, and *ε* represents a random error. The values *X_i_* and *X_j_* correspond to the response of CV dye removal and coded values for the independent factors (−1, 0, and +1). In Equation (1), a positive sign implies that the variables have a synergistic effect, whereas a negative sign indicates that they have an antagonistic effect.

To evaluate the precision of the model, the researchers employed Analysis of Variance (ANOVA) to examine the coefficients. The ANOVA provided several parameters, including *p*-value, *F*-value, determination coefficient (*R*^2^), projected determination coefficient (*R*^2^_pred_), adjusted determination coefficient (*R*^2^_adj_), acceptable precision, degree of freedom (*d*_f_), and standard deviation (SD). The experimental data and model precision were evaluated using these parameters [39]. The researchers used a dependable second-order quadratic model equation to predict the optimal value and describe the interactions between the elements. To determine the factors’ best values, we solved the regression equation, evaluated the counter-response surface map, and set limitations for the variable levels. To establish the variables’ extreme values, preliminary tests have been performed.

### 2.6. Batch Adsorption Studies

To investigate the adsorption capacity of Kaol/Cel for removal of CV dye, a BBD experimental was employed using varying amounts (0–50%) of the adsorbent with 100 mL of dye solution at concentrations of 50–300 mg/L. The experiments were conducted at different temperature intervals (30–60 °C) and pH values ranging from 4 to 10, with adjustments made using 0.1 N HCl and NaOH solutions at different intervals (5–120 min), as detailed in Table 2. Following the adsorption experiments, the residual concentrations were determined using a Cary Series UV-vis spectrophotometer at *λ*_max_ = 590 nm after samples were spun at 3400 rpm for 10 min. The adsorption capacity of Kaol/Cel for CV dye removal (*q*_e_; mg/g) and the percentage of dye removal (*R*%) were calculated using Equations (2) and (3), respectively.
(2)$$R\;\left(\%\right)=\frac{C_{0}-C_{e}}{C_{0}}\times 100$$
(3)$$q_{e}=\frac{V}{W}\left(C_{0}-C_{e}\right)$$
where: *C*_0_ (mg/L) and *C*_e_ (mg/L) are the initial and equilibrium CV dye concentrations, respectively, *V* (L) is the dye solution’s volume, and *W* is Kaol/Cel weight in grams (g).

## 3. Results and Discussion

### 3.1. Adsorbent Characterization

XRD patterns of initial cellulose are shown in Figure 2a–c. For the XRD pattern of kaolinite (a), the well-known diffraction peaks at 2θ around 6.0°, 19.6°, 20.8° 26.5°and 29.3° were attributed to the planes of (110), (002), (020) and (021), respectively, all of which were in agreement with the different characteristic orientation of the layers in the crystal structure of kaolinite (Kaol) [19]. For the XRD pattern of cellulose (b), the peaks observed at 2θ around 5.7°, 14.1°, 20.8°, 27.0°, and 29.0° are attributed to the (101), (10-1), (002), (110), and (10-2) planes, respectively, in the cellulose structure. These peak positions represent the interplanar spacing and arrangement of the cellulose chains [40]. The XRD pattern of the Kaol/Cel–25 composite(c) shows a significant decrease in peak intensities compared to the individual kaolin and cellulose patterns. This decrease in density is due to integrating the cellulose molecules into the clay framework. In addition, this work provided further evidence that Kaol particles may be successfully synthesized in the Kaol/Cel–25 formulation [19,40].

FT-IR analysis was employed to compare the functional groups on the surface of Kaol/Cel–25 before and after CV adsorption, as illustrated in Figure 3a,b, respectively. The FT-IR spectra of Kaol/Cel–25 before CV adsorption (Figure 3a) showed bands at 1633 cm^−1^, which could be attributed to O-H stretching and coordinated water bending vibrations, respectively [16,41]. The Si-O-Si stretching vibrations of kaolinite or quartz cause the peak at 803 cm^−1^, whereas the peaks at 3625 cm^−1^ are caused by Al-OH-Al and Fe-OH-Al deformation. The spectrum of the utilized cellulose was utilized to determine the various chemical groups of cellulose, such as β-glycosidic linkages at approximately 987 cm^−1^, -C-O-C pyranose rings at around 1630 cm^−1^, -C-H groups at around 995 cm^−1^ as well as approximately 2157 cm^−1^ [42]. The FT-IR spectrum of Kaol/Cel–25 after CV dye adsorption (Figure 3b) displayed the same bands in the spectrum before, with a slight shifting of some bands, indicating the involvement of the functional groups of Kaol in the adsorption of CV dye. Moreover, a new band that appeared at the approximately 1744 cm^−1^ band is attributed to the elongation of the C=N bond. The 1582 cm^−1^ band is due to aromatic C–C stretches, while the vibration of aromatic tertiary amine N–C is observed at 1366 cm^−1^, indicating that the important functional groups of Kaol/Cel–25 were responsible for the adsorption of CV dye [43].

Figure 4a depicts the surface morphology of Kaol, which may be described as an uneven and heterogonous surface with crevices. These fissures can be seen across the surface. The EDX spectrum of Kaol reveals the presence of O, Al, Si, K, Fe, and Ti. These elements are present in various minerals such as kaolinite, quartz, and other clay constituents, as identified through XRD analysis. Before CV dye adsorption, the result of the SEM-EDX analysis for Kaol/Cel can be shown in Figure 4b. The appearance of the surface morphology is a surface with protrusions of varying widths with many big spaces and fissures. The EDX analysis verifies the existence of the components O, C, Al, Si, K, Fe, and Ca. This suggests that the clay matrix effectively incorporated the cellulose particles.

Nevertheless, the morphological structure of Kaol/Cel–25 following CV dye absorption (Figure 4c) was converted into a different state, resulting in a reduced number of many tiny apertures and surface slits of the material. This observation provides evidence that molecules of CV dye have been loaded onto the surface of Kaol/Cel–25. In addition, the EDX examination reveals a spike in the carbonation rate in the related EDX spectrum, which provides more evidence that CV is being adsorbed to the Kaol/Cel–25 surface.

### 3.2. Fitting the Process Models

As was indicated before, BBD was responsible for the design of forty-six (46) different experiments (Table 2). The BBD method was utilized to explore the individual and interaction impacts that each of the examined factors had on the CV removal efficiency (as a response). The examined factors were the temperature (D), the pH (C), the adsorbent dose (A), and the temperature (D). These four factors were considered to be independent process factors.

The quadratic polynomial model established the mathematical relationship between the process components and the response. The regression model’s significance test for the response was conducted, and the findings of the ANOVA are presented in Table 3. The model’s *F*-value is 5.66, and the *p*-value is less than 0.0001, indicating the significance of the model terms. A *p*-value of < 0.05 is seen as important under the selected circumstances. The essential model terms for the response, i.e., the elimination of CV, were identified as A, B, BC, BD, and A^2^. However, it was observed that the residence period factor (E) had no effect on the CV elimination, possibly due to the adsorption process’s low sensitivity to time changes. The empirical relationship between CV elimination and the significant variables is expressed by the quadratic regression model given in Equation (4).


CV removal (%) = +97.97 + 11.83A + 5.89B − 8.95BC + 6.53BD − 9.66A^2^(4)


**Table 3 materials-16-04082-t003:** Analysis of variance (ANOVA) of the crystal violet (CV) dye removal response surface quadratic model (*d*_f_: degree of freedom).

Source	Sum of Squares	*d* _f_	MeanSquare	*F*-Value	*p*-Value	Remarks
Model	4461.30	20	223.07	5.66	<0.0001	Significant
A: Cel loading	2242.11	1	2242.11	56.92	<0.0001	Significant
B: Adsorbent dose	555.69	1	555.69	14.11	0.0009	Significant
C-pH	106.31	1	106.31	2.70	0.1130	Insignificant
D-Temp.	7.19	1	7.19	0.1826	0.6728	Insignificant
E-Time	2.57	1	2.57	0.0652	0.8006	Insignificant
AB	0.0015	1	0.0015	0.0000	0.9951	Insignificant
AC	3.04	1	3.04	0.0772	0.7834	Insignificant
AD	49.21	1	49.21	1.25	0.2743	Insignificant
AE	26.74	1	26.74	0.6787	0.4178	Insignificant
BC	321.01	1	321.01	8.15	0.0085	Significant
BD	170.74	1	170.74	4.33	0.0477	Significant
BE	0.9046	1	0.9046	0.0230	0.8808	Insignificant
CD	0.7121	1	0.7121	0.0181	0.8941	Insignificant
CE	0.8080	1	0.8080	0.0205	0.8873	Insignificant
DE	1.85	1	1.85	0.0469	0.8303	Insignificant
A^2^	814.98	1	814.98	20.69	0.0001	Significant
B^2^	166.50	1	166.50	4.23	0.0504	Insignificant
C^2^	3.48	1	3.48	0.0882	0.7689	Insignificant
D^2^	11.12	1	11.12	0.2823	0.5999	Insignificant
E^2^	0.1796	1	0.1796	0.0046	0.9467	Insignificant
Residual	984.81	25	39.39			
Cor Total	5446.11	45				

A positive sign in Equation (4) indicates that the factor has a synergistic impact, while a negative sign means that the factor has an antagonistic effect [44]. As shown in Figure 5, the R-squared (determination coefficient) for the response factors is 0.97, which is a very high number indicating an excellent correlation between the real and projected values.

### 3.3. Interactions Significant for Crystal Violet (CV) Dye Removal

The influence of the adsorbent dose (B) and the solution pH (C) on the CV elimination efficacy showed a significant interaction (as evidenced by the *p*-value of 0.0085 from Table 3). The remaining independent variables (i.e., Cel loading (Kaol/Cel–25), residence period of 5 min, and temperature of 45 °C), were held constant during the experimentation range. Figure 6a,b depict the 3D surface and 2D contour plots for the BC interaction, respectively.

Figure 5 depicts that augmenting the maximum adsorbent injection from 0.02 g to 0.05 g and decreasing the solution pH from 10 to 4 significantly augmented CV elimination efficacy from 61.82% to 99.86%. The greatest CV removal efficiency was achieved at pH 4, with a gradual decrease in dye removal as the pH value increased toward an alkaline environment. Furthermore, Figure 7 shows that the pH_pzc_ of Kaol/Cel–25 was 8.71, establishing that the surface of Kaol/Cel–25 begins to be positively charged at pH values less than pH_pzc_. Conversely, at pH values above pH_pzc_, the surface charge of Kaol/Cel–25 becomes negative, which implies that Kaol/Cel–25 may adsorb cationic dyes. Consequently, more essential electrostatic attractions arise between the surface functional groups of negatively charged Kaol/Cel–25 and cationic CV dye, as shown in Equation (5).

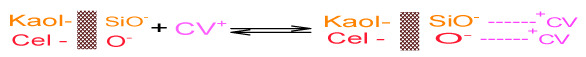
(5)

Table 3 was used to assess the significance of the interactions between the independent variables. The results showed that the interaction influence between adsorbent dosage (B) and temperature (D) had a statistically considerable impact on CV removal efficiency, with a *p*-value of 0.0477. The other variables, namely a 25% Cel loading, a pH of 10, and a 5 min contact time, remained constant throughout the experiment. The interaction between the adsorbent dose and the temperature was further analyzed using 3D and 2D response surface plots, as shown in Figure 8a,b, respectively. The plots indicated that augmenting the adsorbent dose from 0.02 g to 0.05 g led to higher CV dye removal (%), possibly due to increased active adsorption sites or available surface area. Nevertheless, the temperature seemed to possess no significant influence on eliminating CV dye, suggesting an exothermic adsorption process. These results will be discussed further in Section 3.7 on Adsorption Thermodynamics.

It was determined whether there was a statistically crucial interaction between each pair of independent variables (as shown in Table 3). The results revealed that the interaction influence between adsorbent dosage (B) and temperature (D) had a significant impact on the CV removal efficiency (*p*-value = 0.0477). The other independent factors, including Cel loading of 25%, solution pH of 10, and contact time of 5 min, remained constant throughout the experiment. Figure 8a,b illustrate the response surface plots for the interaction between temperature and adsorbent injection. The findings show that augmenting the adsorbent injection from 0.02 g to 0.05 g resulted in higher removal of CV, which can be related to an elevation in active adsorption sites and/or surface area. Moreover, the results suggest that the adsorption of dye molecules onto the Kaol/Cel–25 surface was exothermic since the temperature had no discernible influence on CV removal effectiveness. A more detailed discussion on adsorption thermodynamics is included in Section 3.7.

### 3.4. Adsorption Studies

In this work, the influence of residence period vs. initial CV level on the adsorption potential of Kaol/Cel–25 was explored over a range of initial levels from 50 to 300 mg/L. Figure 9 shows the experimental results, while the optimal conditions for CV dye removal were determined based on the highest achieved CV dye removal percentage, observed at a constant Kaol/Cel–25 adsorbent injection (0.035 g), pH 10, and 45 °C, as indicated in Run #24 of Table 2. As the initial CV level was augmented, the equilibrium adsorption capacity was also augmented from 51.7 to 297.7 mg/g, as presented in Figure 9. This increase is a result of a greater collision rate between CV dye and the Kaol/Cel–25 surface, resulting from the elevated initial CV concentration. Additionally, the higher concentration gradient facilitated the diffusion of dye molecules into the internal pores of the adsorbent, leading to their migration towards the active adsorption sites, thus enhancing the adsorption capacity [45].

### 3.5. Kinetic Modeling

To comprehend the adsorption pathway and behavior of CV on the surface of Kaol/Cel–25, further investigations were carried out, and understanding the adsorption kinetics is crucial. In this context, two kinetic models, pseudo-first-order (PFO) and pseudo-second-order (PSO), were utilized. These non-linear kinetic models are represented by Equation (6) for PFO [46] and Equation (7) for PSO [47]:(6)$$q_{t}=q_{e}\;\left(1-exp^{-k_{1\;}t}\right)$$
(7)$$q_{t}=\frac{q_{e\;k_{2}}^{2\;}t}{1+q_{e}k_{2}t}$$
designated by *q*_e_ (mg/g). In contrast, the quantity of CV dye adsorbed at a specific time *t* is designated by *q*_t_ (mg/g). Additionally, the rate constant of the PFO is represented by *k*_1_ (1/min), and the rate constant of the PSO is represented by *k*_2_ (g/mg min).

Table 4 displays the correlation coefficients (*R*^2^) and model parameters for the PSO and PFO. On the basis of the kinetic adsorption data in Table 4, it can be shown that the PSO model fits the Kaol/Cel–25 dye adsorption on the Kaol/Cel–25 surface better than the PFO model. This is confirmed by the fact that the PSO model had higher *R*^2^ values than the PFO model. Furthermore, the *q*_e_ (i.e., *q*_e,cal_) values estimated with the PSO model are closer to the experimental *q*_e_ (i.e., *q*_e,exp_) values than the *q*_e_ values calculated with the PFO model. These findings and FT-IR analysis suggest that chemical interaction between CV dye and active functional groups on the Kaol/Cel–25 surface is essential in the adsorption process. In the Kaol/Cel–25 surface, the adsorption of CV dye is thus governed by the chemisorption phenomenon.

### 3.6. Isotherms for Adsorption

The adsorption isotherm is a critical piece of information to have when attempting to describe the interaction between the molecules of CV dye and Kaol/Cel–25. As a result, the isotherms that are applied the most often, namely Langmuir [48], Freundlich [49], and Temkin [50], are selected for the purpose of conducting an analysis of the data pertaining to equilibrium adsorption and determining the adsorption potential of Kaol/Cel–25. The non-linear forms of the Langmuir, Freundlich, and Temkin equations are each represented in their own separate Equations (8)–(10):(8)$$\frac{C_{e}}{q_{e}}=\frac{q_{\max}\;K_{a\;\;\;}C_{e}}{1+K_{a}\;C_{e}}$$
(9)$$q_{e}=K_{f}\;C_{e}^{\frac{1}{n}}$$
(10)$$q_{e}=\frac{RT}{b_{T}}(\ln K_{T}C_{e})$$

The variables used in the analysis include *C*_e_ (mg/L) for the equilibrium concentration of CV dye, *q*_max_ (mg/g) for the maximum quantity of CV dye that can be adsorbed per unit mass of Kaol/Cel–25, and *q*_e_ (mg/g) for the quantity of CV dye absorbed per unit weight. The constants used include Langmuir (*K*_a_), Freundlich (*K*_f_), and Temkin (*K*_T_) constants (L/mg), as well as n for the dimensionless constant indicating adsorption intensity, *b*_T_ (J/mol) for the heat of adsorption, *T* (K) for temperature, and *R* (8.314 J/mol K) for the universal gas constant.

The absorption of CV dye by Kaol/Cel–25 is shown in Table 5 to be best fit by the Langmuir adsorption isotherm model, which has a higher *R*^2^ (0.99) than the Freundlich and Temkin models. This implies that CV dye form a monolayer coverage on the homogeneous surface of the adsorbent that are energetically equivalent [51]. Moreover, using the Langmuir model, we determined that 294.12 mg/g is the *q*_max_ for Kaol/Cel–25. Table 6 compares the *q*_max_ of CV dye adsorption onto Kaol/Cel–25 to that of other adsorbents that have been published for removing CV, highlighting that the cationic dye removal efficiency of Kaol/Cel–25 is highly effective and shows promise for further applications.

### 3.7. Thermodynamic Functions Results

To assess the viability and spontaneity of the CV dye adsorption phenomenon onto the surface of Kaol/Cel–25 and estimate the level of randomness at the interface between the dye and the surface, various adsorption thermodynamic parameters were determined. These parameters include the Gibbs free energy change (Δ*G*°) (kJ/mol), the entropy change (Δ*S*°) (kJ/mol. K) and the enthalpy change (Δ*H*°) (kJ/mol). Such parameters thermodynamic were computed using Equations (11)–(13) [61]:(11)$$∆G=-RTLn\;K_{d}$$
(12)$$K_{d}=\frac{q_{e}}{C_{e}}\;$$
(13)$$LnK_{d}=\frac{\Delta S{}^{\circ}}{\;R}-\frac{\;\Delta H{}^{\circ}}{RT{}^{\circ}}$$

Figure 10 depicts a plot of *lnK_d_* against 1/*T*, from which the thermodynamic parameters (Δ*H*° and Δ*S*°) were calculated. The slope of this plot is Δ*H*°, and the intercept is Δ*S*°.

The results presented in Table 7 show that the Gibbs free energy change (Δ*G*°) for the adsorption of CV dye onto the surface of Kaol/Cel–25 is negative, indicating that the phenomenon is spontaneous [62]. Furthermore, the negative enthalpy values obtained for the adsorption process of CV by Kaol/Cel–25 suggest that the process is exothermic, which is consistent with the results obtained from the BBD parametric optimization depicted in Figure 10. Additionally, the adsorption of CV onto Kaol/Cel–25 appears to cause a greater level of disorder at the interface between the solid and solution, as indicated by the negative entropy values [63].

### 3.8. Mechanisms of Adsorption

A proposed adsorption mechanism for CV by Kaol/Cel–25 is illustrated in Figure 11. The various functional groups existing on the surface of Kaol/Cel–25 participate in different interactions that facilitate the adsorption pathway of CV. Notably, the most impactful interaction is an electrostatic attraction between the CV dye molecules and the surface of the Kaol/Cel–25 adsorbent. Such an adsorption pathway encompasses the electrostatic interaction between CV dye cations and the negatively charged sites on the surface of Kaol/Cel–25. Additionally, there is the possibility of *n*-π interaction, which typically occurs when the lone pair electrons on an O are spread out into the orbital of an aromatic ring of dye [64]. Two kinds of hydrogen bonding take place between the Kaol/Cel–25 and the molecular structure of CV. The first and more frequent kind is dipole–dipole hydrogen bonding interaction between present O on the surface of Kaol/Cel–25 with the O and N atoms existing in the CV molecular structure. This interaction is illustrated in Figure 11. The last kind is Yoshida H-bonding, which takes place between -OH groups on the surface of Kaol/Cel–25 and the CV aromatic ring [65]. Based on the information presented above, it can be inferred that these interactions had a crucial contribution in improving the adsorption phenomenon of CV dye on the surface of Kaol/Cel–25.

## 4. Conclusions

This research aimed to develop an environmentally friendly composite material called kaolinite-cellulose (Kaol/Cel) using waste red bean peels (*Phaseolus vulgaris*) as a cellulose source. A Box–Behnken design was employed to perform parametric optimization of the synthesis conditions and adsorptive performance of the Kaol/Cel composite to remove crystal violet (CV) dye. The composite’s polymeric matrix with uniform Cel distribution was found to contribute to its elevated adsorptive behavior towards CV. The best parameters for CV elimination (99.58%) were identified as 25% Cel loading, 0.035 g adsorbent dosage, pH 10, 45 °C temperature, and 5 min contact time. The equilibrium data were best represented by the Freundlich isotherm model, and the Langmuir model revealed a maximum adsorption capacity (*q*_max_) of 294.12 mg/g at 45 °C. The adsorption mechanism involved several interactions, including electrostatic attractions, *n*-π stacking interactions, dipole–dipole hydrogen bonding interactions, and Yoshida H-bonding. Additionally, the adsorption process of CV by Kaol/Cel–25 was observed to be exothermic and spontaneous based on adsorption thermodynamic functions. These findings suggest that Kaol/Cel–25 could be a cost-effective composite biosorbent for removing cationic dyes from aquatic environments and could have potential applications in wastewater treatment. This refers to the elimination of organic and inorganic pollutants, as well as the diminution of the chemical oxygen demand.

## Figures and Tables

**Figure 1 materials-16-04082-f001:**
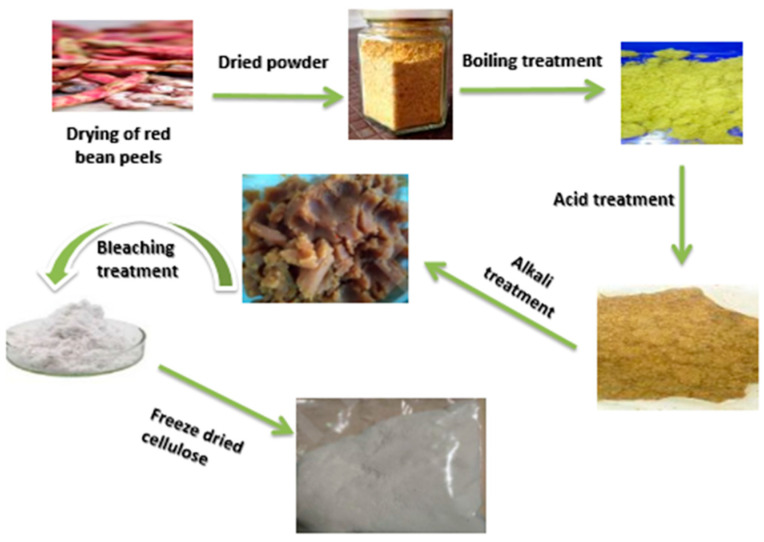
Flow process diagram for cellulose extraction from red bean peels (RBPs).

**Figure 2 materials-16-04082-f002:**
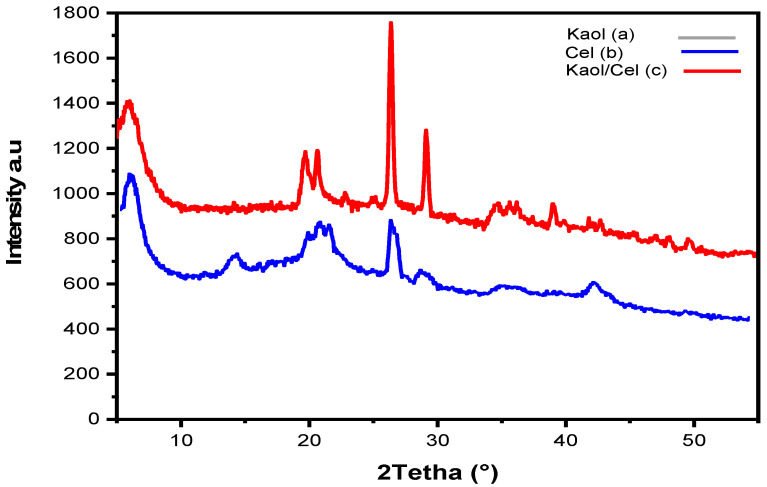
X-ray diffraction (XRD) pattern of (**a**) Kaol, (**b**) Cel, and (**c**) Kaol/Ce–25.

**Figure 3 materials-16-04082-f003:**
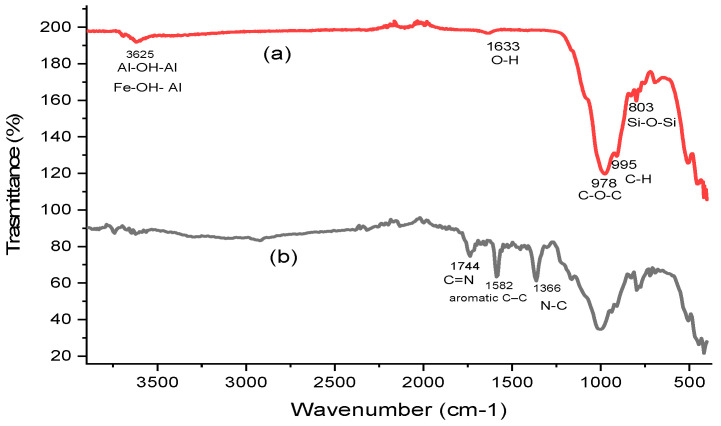
Fourier-transform infrared (FT-IR) spectra of (**a**) Kaol/Cel–25, and (**b**) Kaol/Cel–25 after adsorption of CV dye.

**Figure 4 materials-16-04082-f004:**
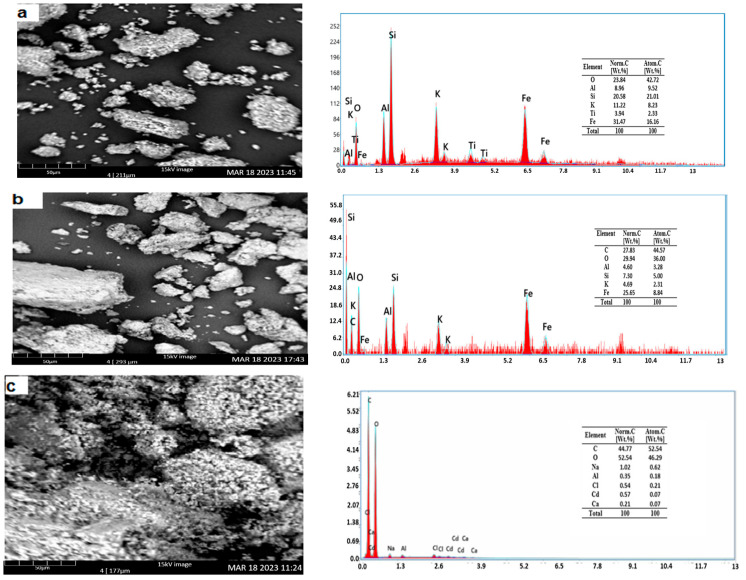
Scanning electron microscopy (SEM) and energy-dispersive X-ray (EDX) spectrum of (**a**) Kaol, (**b**) Kaol/Cel–25, and (**c**) Kaol/Ce–25 after CV dye adsorption.

**Figure 5 materials-16-04082-f005:**
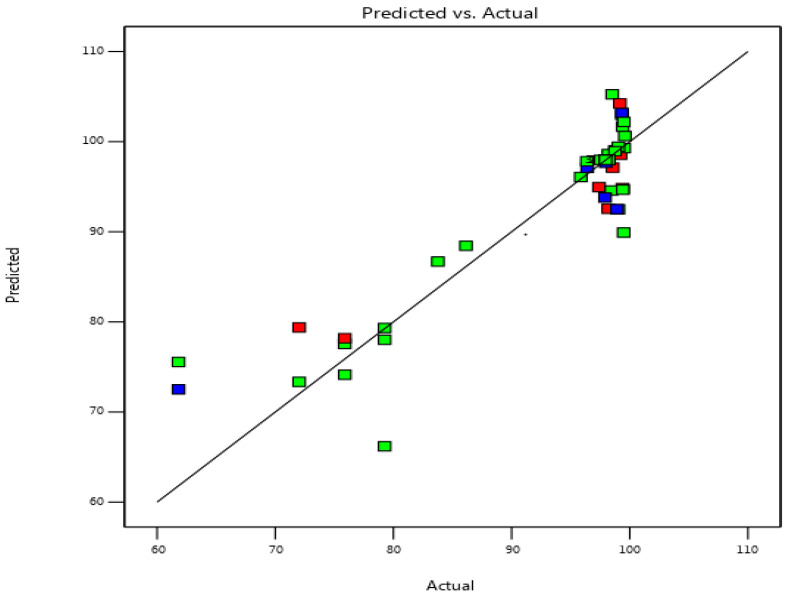
Linear correlation between predicted values vs. the observed values of CV adsorption on Kaol/Cel–25.

**Figure 6 materials-16-04082-f006:**
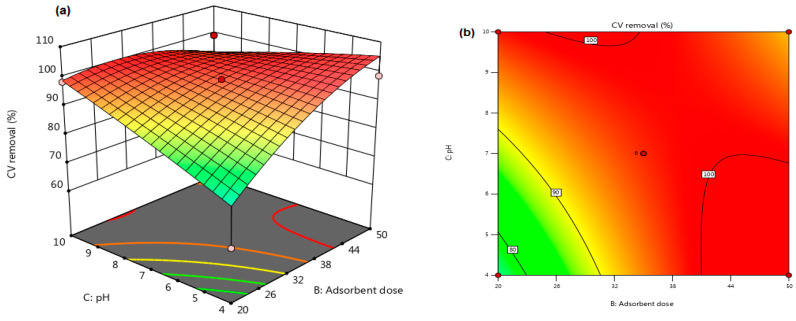
(**a**) 3D surface, (**b**) 2D contour plot of the influence of adsorbent dose and solution pH of CV adsorption on Kaol/Cel–25.

**Figure 7 materials-16-04082-f007:**
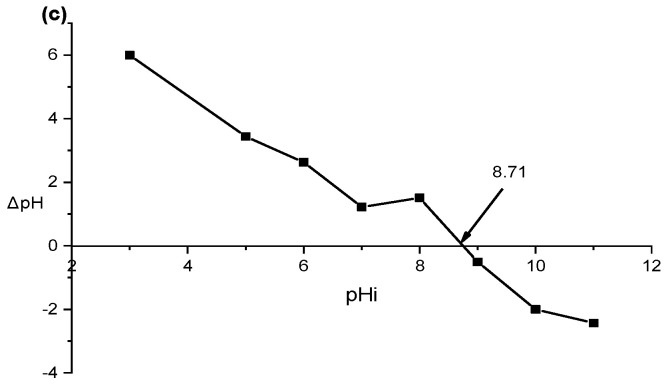
The zero-point of charge (pH_pzc_) of CV adsorption on Kaol/Cel–25.

**Figure 8 materials-16-04082-f008:**
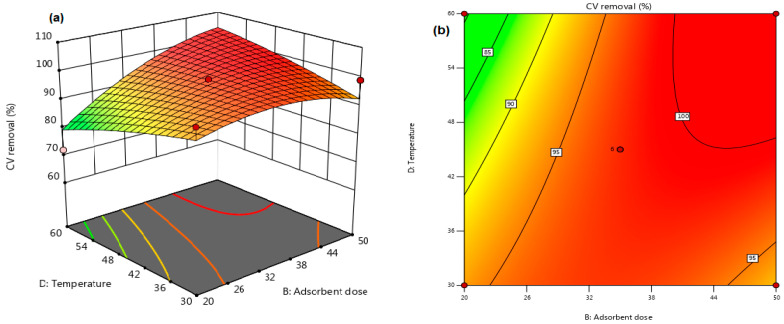
(**a**) 3D surface, and (**b**) 2D contour plot of the influence of adsorbent dose and temperature on Kaol/Cel–25 CV adsorption.

**Figure 9 materials-16-04082-f009:**
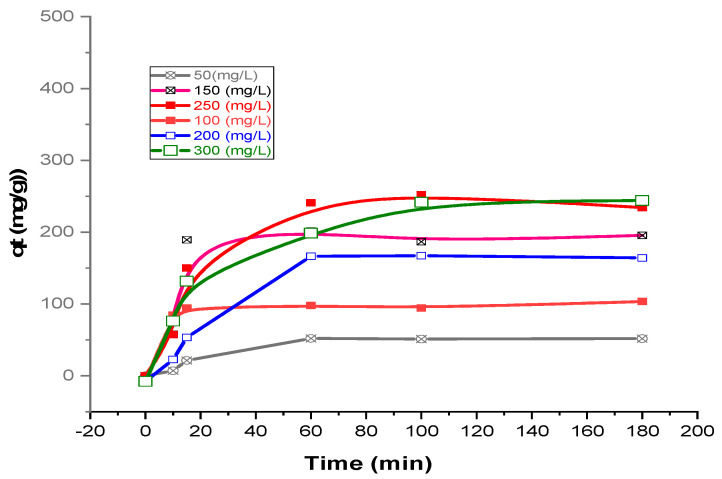
Influence of CV adsorption contact time vs. initial concentrations on Kaol/Cel–25.

**Figure 10 materials-16-04082-f010:**
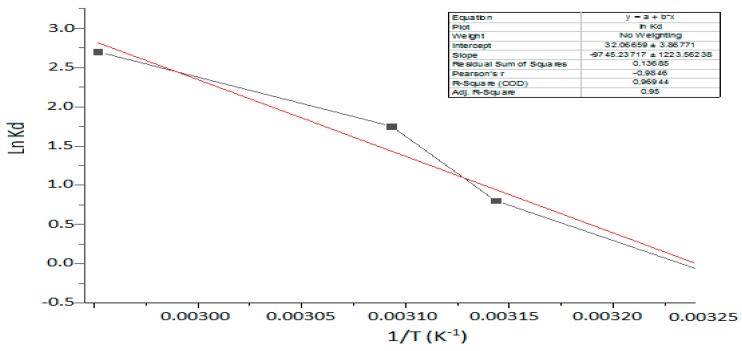
Plot of Van’t Hoff equation for CV adsorption onto Kaol/Cel–25.

**Figure 11 materials-16-04082-f011:**
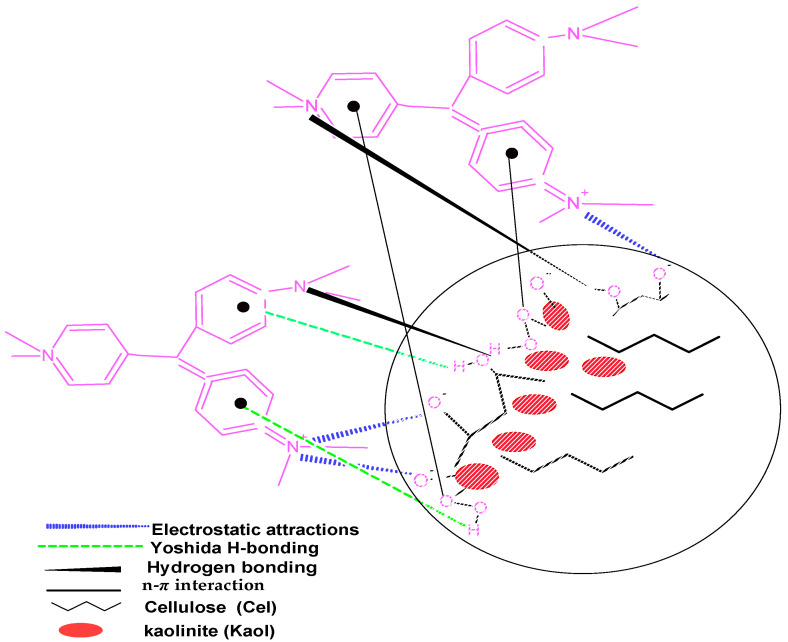
Possible mechanism between Kaol/Cel–25 surface and crystal violet (CV) dye.

**Table 1 materials-16-04082-t001:** Box–Behnken design (BBD) codes for independent factor experimental levels.

Factors	Levels		
	Low (−1)	Medium (0)	High (+1)
A: Loading (%)	0	25	50
B: Adsorbent dose (g)	0.02	0.035	0.05
C: pH	4	7	10
D: Temperature (°C)	30	45	60
E: Contact time (min)	5	65.5	120

**Table 2 materials-16-04082-t002:** Box–Behnken design (BBD) matrix with five factors and experimental results for crystal violet (CV).

Run	A: Cel Loading (%)	B: Adsorbent Dose (g)	C: pH	D: Temperature (°C)	E: Contact Time (min)	Dye Removal (%)
**1**	25	0.035	7	45	17.5	97.95
**2**	0	0.035	7	60	17.5	75.88
**3**	25	0.020	*7*	*60*	17.5	72.02
**4**	25	0.050	10	45	17.5	99.05
**5**	25	0.035	10	30	17.5	98.75
**6**	0	0.035	7	45	30	72.02
**7**	50	0.035	10	45	17.5	99.34
**8**	50	0.020	*7*	*45*	17.5	99.48
**9**	0	0.050	7	45	17.5	79.25
**10**	25	0.020	*4*	*45*	17.5	61.82
**11**	25	0.050	7	45	30	99.05
**12**	25	0.050	*7*	*60*	17.5	99.16
**13**	25	0.050	7	30	17.5	98.91
**14**	25	0.035	7	45	17.5	97.95
**15**	50	0.035	7	45	5	96.37
**16**	25	0.020	10	45	17.5	98.19
**17**	25	0.035	10	60	17.5	99.21
**18**	25	0.035	7	60	5	98.55
**19**	25	0.035	7	45	17.5	97.95
**20**	50	0.035	7	45	30	99.49
**21**	50	0.035	7	60	17.5	99.37
**22**	0	0.035	7	30	17.5	61.82
**23**	25	0.035	10	45	5	99.58
**24**	25	0.050	4	45	17.5	98.51
**25**	25	0.035	4	60	17.5	98.18
**26**	0	0.035	4	45	17.5	75.88
**27**	25	0.020	7	45	5	86.14
**28**	25	0.020	7	30	17.5	97.91
**29**	25	0.035	7	60	30	97.40
**30**	25	0.050	7	45	5	99.52
**31**	0	0.035	10	45	17.5	75.88
**32**	25	0.035	7	30	30	97.97
**33**	50	0.035	7	30	17.5	99.34
**34**	25	0.035	4	45	30	99.41
**35**	0	0.035	7	45	5	79.25
**36**	25	0.035	4	30	17.5	99.41
**37**	25	0.035	4	45	5	98.44
**38**	25	0.020	7	45	30	83.76
**39**	25	0.035	7	45	17.5	98.22
**40**	50	0.035	4	45	17.5	95.86
**41**	25	0.035	7	45	17.5	97.54
**42**	25	0.035	7	45	17.5	98.22
**43**	25	0.035	7	30	5	96.40
**44**	50	0.050	7	45	17.5	99.40
**45**	0	0.020	*7*	45	17.5	79.25
**46**	25	0.035	10	45	30	98.75

**Table 4 materials-16-04082-t004:** The kinetic parameters of pseudo-first-order (PFO) and pseudo-second-order (PSO) for crystal violet (CV) sorption onto Kaol/Cel–25 at optimal conditions.

Concentration (mg/L)	*q*_e,exp_ (mg/g)	PFO			PSO		
		*q*_e_,_cal_ (mg/g)	*k*_1_ (1/min)	*R* ^2^	*q*_e_,_cal_ (mg/g)	*k*_2_ 10^−2^ (g/mg. min)	*R* ^2^
50	51.74	54.23	0.0459	0.963	59.52	0.074	0.994
100	131.5	130.53	0.0531	0.955	133.33	0.298	0.999
150	195.6	180.33	0.0277	0.994	198.33	0.042	1
200	203.2	182.66	0.0215	0.945	208.56	0.016	0.997
250	252.1	192.31	0.0318	0.938	294.12	0.014	0.995
300	297.7	274.82	0.0434	0.954	299.58	0.020	0.996

**Table 5 materials-16-04082-t005:** Langmuir, Freundlich, and Temkin constants for the adsorption of crystal violet (CV) dye onto Kaol/Cel–25 at 45 °C (318.15 K).

Model	Parameters	Value
**Langmuir**	*q*_max_ (mg/g)	294.12
	*K*_a_ (L/mg)	0.03
	*R* ^2^	0.99
**Freundlich**	*K*_f_ (mg/g) (L/mg)^1/n^	38.37
	*n*	2.43
	*R* ^2^	0.98
**Temkin**	*K*_T_ (L/mg)	0.07
	*b*_T_ (JHZ[J/mol])	22.40
	*R* ^2^	0.91

**Table 6 materials-16-04082-t006:** Comparison of the *q*_max_ (mg/g) values for crystal violet (CV) dye adsorption to that of various adsorbents.

Adsorbents	*q*_max_ (mg/g)	References
Palm kernel fiber	78.9	[52]
Rice husk NaOH-modified	44.87	[53]
Fly ash	74.6	[54]
Cellulose-based from sugercane bagasse	107.5	[55]
Magnetite graphene oxide-doped super adsorbent hydrogel	88.78	[56]
Rubber seed pericarp treated with sulfuric acid	302.7	[57]
Microalgae	243.0	[58]
Zeolite–montmorillonite	150.52	[59]
Durian seeds powder	158	[60]
Kaol/Cel–25	294.12	This study

**Table 7 materials-16-04082-t007:** Parameters thermodynamic of crystal violet (CV) dye adsorption onto Kaol/Cel–25.

*T* (K)	*Lnk_d_*	Δ*G°* (kJ/mol)	Δ*H°* (kJ/mol)	Δ*S°* (kJ/mol K)
303.15	0.0867	−0.22	−78.65	−0.238
313.15	0.5931	−1.54		
318.15	1.5411	−4.07		
333.15	2.9099	−8.06		

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
