# Peer review of "Modeling and Optimizing the Crystal Violet Dye Adsorption on Kaolinite Mixed with Cellulose Waste Red Bean Peels: Insights into the Kinetic, Isothermal, Thermodynamic, and Mechanistic Study"

_materials, 2023, doi:10.3390/ma16114082_

Round 1

Reviewer 1 Report

The manuscript is interesting, based on utilizing local natural resources to solve the demanding problem of water contamination. However, the manuscript must be extensively corrected before acceptance. I am sending some comments to be addressed:

1.      The second sentence in the abstract has no meaning in its position, please rewrite

2.      The introduction is full of redundancy and needs to be carefully revised and modified, the same as the rest of the text.

3.      Red bean is a common name for various types of seeds, what is the specific type (scientific name) of red beans used during the study? What do you mean by peels, external pods’ shells or hull?

4.      Step (iii), line 139, not clear, modify it., Also, the procedures for the pretreatment of raw clay is confusing in many steps, please rewrite in clear language.

5.      Line 158: “by mixing it twice with a 1 M HCl solution„ What do you mean?

6.      What did you use silver nitrate (mentioned in the chemicals list) for?

7.      Section 2.4, what “bentonite„ is doing in the preparation of Kaol/Cel composite

8.      “10 grams of bentonite was mixed with 16 mL of 46% NaOH in ice water and 170 magnetic stirring for six hours„ 10 g of clay in 16 mL, explain

9.      The preparation procedures of Kaol/Cel composites must be rewritten in clear language.

10.  Please add the XRD pattern of the Kaolinite and cellulose individuals before modification

11.  The FT-IR spectroscopic analysis part must be revised, and peak positions need to be specified on the figure and assigned correctly.

12.  The image resolution of SEM analysis is very poor to study the morphology of the prepared composites, and the descriptions on the text do not belong to what somebody can see on the images.

13.  The EDX analysis of the composites before and after the CV adsorption prove the presence of Cl with almost the same content, hence the conclusion of “the EDX examination reveals the presence of Cl, which provides more evidence that CV is being adsorbed to the 271 Kaol/Cel-25 surface„ is incorrect.

14.  Figure 8 shows a fast adsorption process takes place in less than 10 minutes, such current data cannot lead to a correct prediction of kinetic modeling, smaller time intervals at the very beginning of the adsorption should be considered to fellow the adsorption process and kinetic modeling should be reevaluated. Figure 8 shows also, that the capacity of the adsorbent can be predicted to be still much higher than 476 mg/g (mentioned as the maximum adsorption capacity, if we consider the correct calculations of dye concentration, and adsorption capacity Qt.

15-ordering and style of the references need to be fixed.

English should be carefully checked, specially sentences that are not completed, or jumping from one idea to another with no meaning. 

Author Response

Thank you to see the attached file

Reviewer 2 Report

The manuscript is suitable for publication since it contains a well conducted study that is well written.

However, it can be improved by the increase of the quality of figures : - for all figures, the character size is too small ; - in Fig2 of XRD, authors should add the identification of main peaks and the pattern must be shortened above 10° 2teta ; - Fig6 is too complex and should be separated into 2 figures ; - the colors and the character sizes in figure 10 are not suitable, and authors must improve it to enhance the discussion section.

Besides, Table 2 is too large and it is suggested to reduce the number of data to the interval used. I also suggest to made some corrections in the reference section.

Author Response

Thank you to see the attached file

Reviewer 3 Report

I reccomend the paper for pubblication, after the modification in the attached document.

·         abstract is pretty confusing I would re-write it. For Instance, I would eliminate the detailed description of the investigated parameters in the abstract. 

·         “After that, the fractions were transferred to a cylinder of glass (1000 mL) were added to dissolve the carbonate salts acetate buffer solution of pH 4.8 (80 mL) by to dissolve the carbonate salts” these sentence needs to be re-written, it is quite unclear. Same thing on line 143 “incorporating a dispersible component, namely sodium hexametaphosphate (NaPO3)6”. I’d simply write “disperse with sodium hexametaphosphate (NaPO3)6.

·         Regarding kaolinite Specific Surface Area, would be possible to have the entire adsorption desorption curve and, if there are any, pore size distribution?

·         About cellulose extraction “The filtered residue was then subjected to bleach treatment twice with a  4% (w/v) NaOCl solution (500 mL) for 60 min at 80-90 °C, resulting in white-colored cellulose”, was this step performed at a buffered pH to avoid Cl2 dismutation?

·         Line 170: “Initially, 10 grams of bentonite was mixed with 16 mL of 46% NaOH in ice water and 170 magnetic stirring for six hours”. Was it bentonite or modified kaolinite?

·         Line 216: “Following the adsorption experiments, the residual concentrations were determined using a Cary Series UV-vis spectrophotometer at λmax = 590 nm after 217 samples were spun at 3400 rpm for 10 min”. Was the adsorbent removed from the solution before performing Uv-vis experiments?

·         Line 232: “These distinct peaks elucidate the structure of Kaol/Cel-25, which identifies the crystalline structure of Kaol and the intramolecular hydrogen bonding between Cel and Kao”. Isn’t INTERMOLECULAR hydrogen bonds?

·         Was CV adsorption capacity also determined individually for kaolinite and cellulose? That is, what are the adsorption capacities of the single components of the composite material? Is there any improvement compared to pristine Cel and Kao?

English needs to be revised, although not extensively

Author Response

Thank you to see the attached file

Round 2

Reviewer 1 Report

The manuscript has been improved, however, there are still critical comments must be addressed, I repeat them:

- The preparation procedure of  Kaol/Cel composite not clear still, how could you mix 10 grams of clay (very big amount) with only 16 ml of NaOH solution, to prepare slurry or what, I can't understand. Please clear.

- Figure 2., poor quality of the plots, captions and lagends need to be fixed.

- Figure 4., very poor SEM images, and scale bar is invisible.

- Figure 9 is a big question, the immediate adsorption of all the dye within 10 min, based on those results no kinetics can be predicts, the experiment must be redesigned, by changing the weight of the adsorbents, dye concentration and volume, etc.   

English need to be double checked.

Author Response

Dear Reviewer,

Thank your valuable comments and thank you to see the attached file
Best regards

---------------------------------------------------------------------------------------
Dr. Noureddine Ali Elboughdiri,
 Associate Professor.
 Head of Dept. of Chem. Eng.,
 Dept. of Chem. Eng., College of Eng., Univ. of Hail, Hail, KSA
 Mobile: ++966 549 571 015
---------------------------------------------------------------------------------------
 National School of Engineering Gabes, Gabes 6011, Tunisia
---------------------------------------------------------------------------------------
